# In Vitro Antibacterial Activity of Ceftobiprole and Comparator Compounds against Nation-Wide Bloodstream Isolates and Different Sequence Types of MRSA

**DOI:** 10.3390/antibiotics13020165

**Published:** 2024-02-07

**Authors:** Lingqin Li, Wangxiao Zhou, Yunbo Chen, Ping Shen, Yonghong Xiao

**Affiliations:** 1State Key Laboratory for Diagnosis and Treatment of Infectious Disease, The First Affiliated Hospital, Zhejiang University School of Medicine, Hangzhou 310003, China; slyy_01821@tzc.edu.cn (L.L.); 12018220@zju.edu.cn (W.Z.); 1507115@zju.edu.cn (Y.C.); shenping@zju.edu.cn (P.S.); 2Infectious Department, Taizhou Municipal Hospital, Taizhou 318000, China

**Keywords:** ceftobiprole, minimum inhibitory concentration, bloodstream isolates, MRSA, in vitro activity

## Abstract

Bloodstream infections by bacteria, especially multidrug-resistant bacteria, remain a worldwide public health concern. We evaluated the antibacterial activity of ceftobiprole and comparable drugs against different bloodstream isolates and different sequence types of methicillin-resistant *Staphylococcus aureus* (MRSA) in China. We found that MRSA, methicillin-susceptible *Staphylococcus aureus* (MSSA), and methicillin-susceptible coagulase-negative *Staphylococcus* (MSCNS) displayed ceftobiprole sensitivity rates of >95%, which are similar to the rates for linezolid, daptomycin, and vancomycin. Of the tested MRCNS strains, 90.4% were sensitive to ceftobiprole. The sensitivities of ST59, ST398, and ST22 MRSA to ceftobiprole were higher than that of ST239. Ceftobiprole’s MIC_50/90_ value against *Enterococcus faecalis* was 0.25/2 mg/L, whereas *Enterococcus faecium* was completely resistant to this drug. Ceftobiprole exhibited no activity against ESBL-positive Enterobacterales, with resistance rates between 78.6% and 100%. For ESBL-negative *Enterobacterales*, excluding *Klebsiella oxytoca*, the sensitivity to ceftobiprole was comparable to that of ceftazidime, ceftriaxone, and cefepime. The MIC_50/90_ value of ceftobiprole against *Pseudomonas aeruginosa* was 2/16 mg/L, and for *Acinetobacter baumannii*, it was 32/>32 mg/L. Thus, ceftobiprole shows excellent antimicrobial activity against ESBL-negative *Enterobacterales* and *Pseudomonas aeruginosa* (comparable to that of ceftazidime, ceftriaxone, and cefepime); however, it is not effective against ESBL-positive *Enterobacterales* and *Acinetobacter baumannii*. These results provide important information to clinicians.

## 1. Introduction

Bacterial bloodstream infections remain a major cause of the life-threatening condition of sepsis. Studies have demonstrated that one-third of hospital-associated deaths were related to sepsis, and bacterial infections causing sepsis account for 87% of hospitalizations [1]. Xu et al. found that among 1318 patients with positive blood cultures in China, the associated mortality rate was 48.9% [2]. The GBD 2019 Antimicrobial Resistance Collaborators estimated that deaths caused by 33 bacterial pathogens accounted for 13.6% of all global deaths and 56.2% of all sepsis-related deaths in 2019 [3]. Reducing the mortality burden from infections is an urgent global public health priority. Bloodstream infections have arguably been encompassed within the scope of more recent global sepsis initiatives [4]. Appropriate selection of antimicrobial agents is crucial to enhancing the treatment efficacy for bloodstream infections. In the United States, methicillin-resistant *Staphylococcus aureus* (MRSA) accounted for 11% of bloodstream infections, with extended-spectrum β-lactamases (ESBL)-positive *Enterobacterales* and carbapenem-resistant *Enterobacterales* accounting for 7% and 1.3%, respectively [5]. The use of broad-spectrum antibiotics, particularly those with activity against multidrug-resistant bacteria, remains the major treatment strategy for bloodstream infections. Ceftobiprole, a fifth-generation cephalosporin, differs from most β-lactam antibiotics in that it can bind to penicillin-binding protein 2a (PBP2a) in MRSA and displays associated antibacterial activity. Ceftobiprole medocari, the prodrug of ceftobiprole, has been approved by the Food and Drug Administration (FDA) for the treatment of complicated skin and soft-tissue infections, community-acquired pneumonia, and hospital-acquired bacterial pneumonia (excluding ventilator-associated pneumonia). The monitoring of ceftobiprole activity following its introduction in Europe, the United States, Canada, and other countries has indicated its antimicrobial activity against various bacteria including MRSA, *Enterococcus faecalis*, penicillin-resistant *Streptococcus pneumoniae*, ESBL-negative *Enterobacterales*, and *Pseudomonas aeruginosa* [6,7,8,9]. However, studies on clinical isolates from bloodstream infections in China are limited because ceftobiprole medocari has not yet been employed in this region. We, therefore, conducted a large-sample antimicrobial activity study on bloodstream isolates collected from more than 50 hospitals in China, comparing the activity of ceftobiprole and comparator drugs. We subsequently investigated the antimicrobial activity of these drugs against different sequence types (STs) of MRSA to provide guidance for clinical drug use.

## 2. Results

### 2.1. The Antibacterial Activity of Ceftobiprole and Comparator Compounds against Gram-Positive Bacteria

Ceftobiprole exhibited significant antibacterial activity against methicillin-susceptible *Staphylococcus aureus* (MSSA) and methicillin-susceptible coagulase-negative *Staphylococcus* (MSCNS), with MIC_50/90_ values of 0.25/1 mg/L. Ceftobiprole also displayed activity against MRSA strains, with MIC_50/90_ values of 0.5/1 mg/L. The sensitivity rates of all assayed bacteria were above 95.0%, similar to those of linezolid, daptomycin, vancomycin, cotrimoxazole, and levofloxacin. However, methicillin-resistant coagulase-negative *Staphylococcus* (MRCNS) strains had a sensitivity rate of 90.4% to ceftobiprole (with an MIC_50/90_ of 1/2 mg/L), which was lower than that of linezolid, daptomycin, and vancomycin (Table 1).

Ceftobiprole also exhibited strong activity against *Enterococcus faecalis* with MIC_50/90_ values of 0.25/2 mg/L, while the MIC_50_ and MIC_90_ values against *E. faecium* strains were >32 mg/L. All *Enterococci* were sensitive to vancomycin and teicoplanin. All *E. faecalis* strains were sensitive to tigecycline, whereas 1.5% of *E. faecium* strains were not. The sensitivity rates of *E. faecium* strains to daptomycin and linezolid were higher than those of *E. faecalis* (99.5% vs. 92.9%, and 98.2% vs. 95.3%, respectively) (Table 1).

### 2.2. Sensitivity of Different Sequence Types of MRSA to Ceftobiprole and Comparator Compounds

Among the 294 MRSA strains isolated from bloodstream infections in 2019–2020, the main MLST types were ST59 (33.0%), ST5 (15.0%), ST398 (13.0%), ST22 (5.4%), and ST239 (5.4%). All MRSA strains were susceptible to linezolid, vancomycin, and daptomycin. Ceftobiprole’s MIC_50_ values for all STs, except ST5 and ST239, were 0.5 mg/L. Meanwhile, ST5 and ST239 displayed a higher ceftobiprole MIC_50_ of 2 mg/L. Three ST239 MRSA strains were resistant to ceftobiprole, and resistance to ceftobiprole and cotrimoxazole was only found in ST239 MRSA, with resistance rates of 18.8% (3/16). ST239 MRSA also showed the highest resistance rates to rifampicin (56.3%) and levofloxacin (87.5%) among all MLST types. The antimicrobial activity of all agents against MRSA types other than ST239 was comparable, with the exception of detected resistance to rifampicin and levofloxacin. ST5 MRSA showed higher resistance rates to levofloxacin, but no resistance was observed for the other drugs. The ST22 MRSA strains were sensitive to all the seven drugs tested (Table 2 and Figure 1).

### 2.3. The Antibacterial Activity of Ceftobiprole and Comparator Compounds against Gram-Negative Bacteria 

The sensitivity rates of ESBL-negative *Escherichia coli*, *Enterobacter aerogenes*, and *Klebsiella pneumoniae* to ceftobiprole were 92.1%, 95.9%, and 87.2%, respectively, and these rates were similar to those observed for ceftazidime, ceftriaxone, and cefepime, and slightly lower than that for ertapenem. However, the sensitivity rates of ESBL-negative *Klebsiella oxytoca* were 63.7%, significantly lower than those for the comparator compounds. More than 50.0% of *Enterobacter cloacae*, *Salmonella* spp., and *Serratia marcescens* isolates were inhibited by ceftobiprole at concentrations of 0.06–0.125 mg/L, with sensitivity rates of 74.0%, 88.8%, and 86.0%, respectively, similar to those of ceftriaxone, cefepime, and ceftazidime. For ceftobiprole, almost no activity was observed against ESBL-positive *Enterobacterales*, with resistance rates ranging from 78.6% to 100.0% (Table 3).

Ceftobiprole’s MIC_50/90_ values against *Pseudomonas aeruginosa* were 2/16 mg/L, similar to cefepime (with an MIC_50/90_ of 4/16 mg/L) and slightly better than ceftazidime (with an MIC_50/90_ of 4/32 mg/L). However, this compound showed poor activity against *Acinetobacter baumannii*, with MIC_50/90_ values of 32/>32 mg/L, similar to those of ceftazidime and cefepime (the MIC_50/90_ values of both were 64/>64 mg/L) (Table 3).

### 2.4. The Cumulative MIC Distribution of Ceftobiprole and Comparator Compounds against Major Bacteria

Ceftobiprole was highly potent against *Staphylococci*, with an activity curve shifted to the left of vancomycin, linezolid, cotrimoxazole, and levofloxacin. Its antibacterial activity was also superior to that of daptomycin against MSSA and MSCNS. Daptomycin displayed stronger antibacterial activity than the comparator compounds against MRCNS (Figure 2). The bacteriostatic curves of ceftobiprole, cefepime, ceftriaxone, and ceftazidime against ESBL-positive *E. coli* and *K. pneumoniae* were similar, inhibiting over 90% of bacteria at concentrations of ≥32 mg/L (Figure 3a,c). Ertapenem displayed the greatest activity against ESBL-negative *E. coli* and *K. pneumoniae*, followed by ceftobiprole, cefepime, ceftriaxone, and ceftazidime. Ceftobiprole inhibited over 80% of *P. aeruginosa* at concentrations of ≤4 mg/L. Furthermore, the bacteriostatic curve of ceftobiprole was shifted to the left of cefepime and ceftazidime, with the highest MIC_90_ (32 mg/L) observed for ceftazidime (Figure 3e).

## 3. Discussion

Antimicrobial resistance (AMR) is firmly lodged in the medical community’s collective mind as a challenge to be reckoned with in the decades to come [10]. This study determined the antibacterial activity of ceftobiprole and comparator compounds against 9860 bacterial strains isolated from blood and 294 MRSA strains collected in China. In the period from January 2021 to December 2021, this research involved gathering 9860 strains to assess the in vitro antibacterial properties of ceftobiprole in comparison with other compounds against a range of Gram-positive and Gram-negative bacteria. Moreover, for a detailed investigation into the sensitivity of different MRSA sequence types, 294 MRSA isolates collected between January 2019 and December 2020 were analyzed to determine their reactivity to ceftobiprole and various comparator compounds. We found that ceftobiprole exhibited antibacterial activity against most Gram-positive and Gram-negative bacteria, with most showing activity comparable to or greater than that of commonly used antibacterial drugs.

*S. aureus* was identified as the primary bacterial cause of death in 135 countries and additionally held the highest age-standardized mortality rate in 16 of the 21 GBD regions [3]. This is largely attributed to the emergence of resistant strains, particularly the widespread prevalence of MRSA. The resistance mechanism of MRSA to antibiotics is primarily characterized by alterations in the target site or a reduction in the affinity between the target and the antibiotic. Moreover, changes in cell membrane proteins and plasmid-mediated efflux pumps also play a significant role in the antibiotic resistance observed in MRSA. In this study, ceftobiprole exhibited potent antibacterial activity against MRSA, MSSA, and MSCNS, comparable to or better than that of linezolid, daptomycin, and vancomycin. Despite this, we identified some MRSA, MSCNS, and MRCNS strains that were resistant to ceftobiprole. These findings are consistent with studies conducted in Europe and China [11,12,13]. For example, Amsler et al. reported the in vitro antibacterial activity of ceftobiprole in the United States, with MIC_50/90_ values of 0.5/2 mg/L, 0.25/0.5 mg/L, 0.5/2 mg/L, and 0.25/0.5 mg/L against MRSA, MSSA, MRCNS, and MSCNS, respectively, and Yin et al. identified values 1/2 mg/L, 0.5/1 mg/L, 1/2 mg/L, and 0.25/0.5 mg/L in a study conducted in China, respectively. Holland et al. conducted a double-blind, non-inferiority trial in 387 patients with *Staphylococcus aureus* bacteremia and found that the efficacy of ceftobiprole was non-inferior to that of daptomycin [14], consistent with the in vitro susceptibility data reported in our study. In this study, we found that the ST59, ST398, and ST22 MRSA strains were more sensitive to ceftobiprole, levofloxacin, rifampin, and cotrimoxazole than ST239 MRSA strains and that ceftobiprole-resistant MRSA strains were uncommon. Three resistant strains were found for ST239 MRSA, with MIC values of 4 mg/L. Zhu et al. previously identified a correlation between ceftobiprole’s MIC and MRSA’s genetic background, with stable ceftobiprole resistance mutations in mecA and other genes only occurring in ST5 MRSA and ST239 MRSA. SCCmec amplification may also occur following ceftobiprole exposure in ST239 MRSA isolates, although this has not been observed in the other lineages [15]. Since 2010, the prevalence of ST59 MRSA in China has been increasing, gradually replacing ST239 and becoming the dominant clone in most Chinese hospitals [16,17], supporting the clinical use of ceftobiprole. In an Italian study, three ST228 strains of ceftobiprole-resistant MRSA were detected, with MIC values of 4 mg/L, before the clonal replacement of ST228 with ST22 [18]. Hawser et al. analyzed the whole-genome sequences of three resistant MRSA strains from European countries and identified two as clonal complex 8 (CC8) and one as CC5, with different mutations in the gene encoding the penicillin-binding protein MecA [19]. Therefore, research on different parts of the genotyping of MRSA and the analysis of MRSA whole-genome sequence data can aid in drug resistance monitoring. The antibacterial activity of ceftobiprole against MRCNS in our study (with a sensitivity rate of 90.4% and an MIC_50/90_ of 1/2 mg/L) was slightly lower than that against other *Staphylococci*, which is consistent with the results reported by Pfaller et al. (with a sensitivity rate of 85.7% and an MIC_50/90_ of 1/4 mg/L) [20], but the mechanism of resistance in MRCNS remains unclear. We also identified 13 strains of *Staphylococcus* resistant to linezolid, including two MRSA, ten MRCNS, and one MSCNS, all of which were sensitive to ceftobiprole. These results are consistent with the findings of Rossolini et al. [6]. Despite the use of ceftobiprole in Europe and other countries over the years, the current resistance rate of MRSA to ceftobiprole remains very low (<2%) [11,21,22,23].

In Europe and the United States, *E. faecalis* is predominantly responsible for enterococcal bloodstream infections [24], whereas in China, *E. faecium* is more prevalent [25]. Concurrently, data from the 20-year global SENTRY antimicrobial surveillance program revealed that the resistance rate of *Enterococcus* isolates to vancomycin was 16% in 1997, and this rate has been steadily increasing [26]. In this study, we demonstrated that ceftobiprole displays potent in vitro activity against *E. faecalis*, with MIC_50/90_ values of 0.25/2 mg/L, but not against *E. faecium*. These data are consistent with a study by Yin et al., who reported ceftobiprole MIC_50/90_ values against *E. faecalis* and *E. faecium* of 0.5/1 mg/L and 32/>32 mg/L, respectively [12]. The antibacterial activity of ceftobiprole against *E. faecalis* is mediated by the inhibition of peptidoglycan cross-linking by the acylation of serine at the active sites of penicillin-binding proteins (PBPs) with a greater affinity for PBP4 [27]. Both the overexpression and substitution of amino acids in low-affinity PBP4 can induce resistance [28]. The weaker activity of ceftobiprole against *E. faecium* correlates with the lower affinity of ceftobiprole to PBP5 [29].

According to the study conducted by Swiss ANRESIS, *E. coli* has been identified as the primary resistant pathogen causing bloodstream infections. In terms of treatment, infections caused by third-generation cephalosporin-resistant *E. coli* pose a significant challenge [30]. Like other cephalosporins, the activity of ceftobiprole against *Enterobacterales* largely depends on the expression of β-lactamases. Ceftobiprole is degraded by extended-spectrum β-lactamases, explaining the lack of antibacterial activity against ESBL-positive *Enterobacterales* in our study, whilst carbapenems retain activity [20,31,32]. Ceftobiprole possesses antibacterial activity against ESBL-negative *Enterobacterales* (except for *K. oxytoca*), *E. cloacae*, *Salmonella* spp., and *S. marcescens*, similar to that of ceftriaxone, ceftazidime, and cefepime, but slightly inferior to carbapenems, such as ertapenem. These findings are consistent with other studies conducted in China and Europe [12,20]. A large-sample study by Pfaller et al. demonstrated sensitivity rates of *Enterobacterales* to ceftobiprole, ceftriaxone, ceftazidime, cefepime, and imipenem of 73.8%, 73.3%, 74.2%, 78.2%, and 96.3%, respectively. The ceftobiprole MIC_50/90_ values against ESBL-negative *E. coli* and *K. pneumoniae* were 0.03/0.06 mg/L, similar to those reported by Yin et al. in China (≤0.06/0.25 mg/L). Many *Enterobacterales* and *P. aeruginosa* encode an inducible AmpC β-lactamase, which is usually expressed at low levels but can be up-regulated in response to cell wall damage or exposure to β-lactam antibiotics [33]. Ceftobiprole and cefepime are stable against AmpC enzymes. In this study, most ESBL-negative *Enterobacterales* and *P. aeruginosa* had lower ceftobiprole and cefepime MIC values than ceftazidime, and an analysis of the bacteriostatic curves indicated greater antibacterial activity than ceftazidime. These two drugs may rapidly penetrate the cytoplasm of Gram-negative bacteria [34]. *P. aeruginosa*, a major cause of hospital-acquired infection, is resistant to multiple antibiotics [35]. When the sensitivity breakpoints were set to 4 mg/L, most *P. aeruginosa* isolates were classified as sensitive to ceftobiprole (>80%, with MIC_50/90_ values of 2/16 mg/L). The sensitivity rates were higher than that in Europe (63.2%) [23], but were consistent with previous data from China, with the same MIC_50/90_ values (2/16 mg/mL) [36], observations which are likely indicative of earlier applications of the drug in European countries. The response of *P. aeruginosa* exposed to ceftobiprole differs from that of bacteria exposed to the other two drugs, being characterized by increased efflux rather than overexpression of AmpC [37]. Ceftobiprole and other β-lactam antibiotics have poor antibacterial activity against *A. baumannii*.

The results of the above studies are based on in vitro susceptibility to address clinical needs and guide clinical treatment. However, it is important to note that discrepancies still exist between in vitro drug susceptibility testing and clinical applications in practice. This gap may be attributed to the incorrect identification of pathogenic bacteria, or a failure to consider pharmacokinetic factors in the treatment process. Monte Carlo simulation (MCS), a statistical modeling method, integrates in vitro antimicrobial susceptibility with pharmacodynamic principles, offering substantial reference value for clinical evaluation and the rational use of antibiotics. By employing the MCS method, clinicians can simulate various scenarios for different patient populations to compare the efficacy of different drugs, dosages, and dosing intervals [38].

## 4. Materials and Methods

### 4.1. Strain Sources

From January 2021 to December 2021, a total of 9860 strains of Gram-positive and Gram-negative bacteria causing bloodstream infections were collected from 51 hospitals (16 provinces) participating in the Blood Bacterial Resistance Investigation Collaborative System (BRICS) (Figure 4). The following bacteria were excluded: single-bottle-culture-positive coagulase-negative Staphylococcus, Bacillus, Streptococcus viridans, Corynebacterium, Propionibacterium, Aerococcus, Micrococcus, Brucella, duplicate strains from the same patient, and contaminated bacteria. All strains were stored in Microbank tubes and stored at −80 °C. BRICS members transmitted clinical isolates quarterly to the central lab at Zhejiang University. After receiving the strains, the central lab re-identified and stored them for sensitivity tests. Two-hundred and ninety-four MRSA strains causing bloodstream infections from BRICS’s 54 hospitals from January 2019 to December 2020 were subjected to ST typing and drug susceptibility testing. Quality control strains included *Staphylococcus aureus* ATCC 29213, *Enterococcus faecalis* ATCC 29212, *Escherichia coli* ATCC 25922 and ATCC35218, *Pseudomonas aeruginosa* ATCC 27853, and *Klebsiella pneumoniae* ATCC 700603.

### 4.2. Antibiotics and Determination of Minimum Inhibitory Concentrations (MICs)

Ceftobiprole (BPR; lot no.: YF-BAL9141-000-211209-01; potency: 91.3%) was provided by Shenzhen Huarun Jiuxin Pharmaceutical Co., Ltd. (Shenzhen, China) Penicillin G (PEN; lot no.: 130437-201707; potency: 94.1%), ceftriaxone (CRO; lot no.: 130480-201504; potency: 98.0%), cefepime (FEP; lot no.: 130524-201404; potency: 84.4%), levofloxacin (LVX; lot no.: 130455-201607; potency: 97.0%), erythromycin (ERY; lot no.: 130307-201417; potency: 93.3%), trimethoprim (TMP; lot no.: 100031-201606; potency: 99.0%), and rifampicin (RIF; lot no.: 130496-201403; potency: 98.8%) were purchased from the National Institutes for Food and Drug Control. Ceftazidime (CAZ; lot no.: J1230A; potency: 94.0%), ciprofloxacin (CIP; lot no.: D1201A; potency: 99.0%), ertapenem (ETP; lot no.: S0802A; potency: 90.0%), and sulfamethoxazole (SMZ; lot no.: 321A026; potency: 99.5%) were purchased from Dalian Melone Pharmaceutical Co., Ltd. (Dalian, China) Sulfamethoxazole combined with trimethoprim is referred to as cotrimoxazole throughout (SXT). Drug susceptibilities were determined using the broth microdilution method recommended according to the Clinical and Laboratory Standards Institute (CLSI) guidelines. Tigecycline (TGC), vancomycin (VAN), linezolid (LNZ), daptomycin (DAP), and teicoplanin (TEC) were obtained from Wenzhou Kangtai Biotechnology Co., Ltd. (Wenzhou, China) (lot no.: DZ1185), and susceptibility to these drugs was determined using the broth dilution method. Ceftobiprole susceptibility breakpoints were discerned from the European Committee on Antimicrobial Susceptibility Testing (EUCAST) (https://eucast.org/clinical_breakpoints/ (accessed on 10 December 2022)), while those of the other drugs were obtained from CLSI standards [39].

### 4.3. MRSA Sequence Typing

The genomic DNA of 294 MRSA strains was extracted using the Ezup Column Bacteria Genomic DNA purification kit (Sangon Biotech, Shanghai, China). The genomes of the selected MRSA strains were sequenced using the Illumina HiSeq X 10-PE150 sequencing platform. The raw sequencing reads were assembled using the Shovill v1.1.0 pipeline (https://github.com/tseemann/shovill (accessed on 5 February 2023)). An online tool was used for MLST typing (https://cge.food.dtu.dk/services/MLST/ (accessed on 11 February 2023)).

### 4.4. Statistical Analysis

WHONET5.6 software was used to statistically analyze the drug susceptibility results.

## 5. Conclusions

Ceftobiprole displayed strong antibacterial activity against *Staphylococci* and *E. faecalis* in vitro, at levels comparable to vancomycin, daptomycin, and linezolid. The clinically dominant MRSA types in China, including ST59, ST398, and ST22, showed higher sensitivity to ceftobiprole, levofloxacin, and rifampicin compared with ST239. Ceftobiprole displayed antibacterial activity against ESBL-negative *Enterobacterales* (except for *K. oxytoca*), *E. cloacae*, *Salmonella* spp., *S. marcescens*, and *P. aeruginosa* in vitro at levels comparable to ceftazidime, ceftriaxone, and cefepime. However, ceftobiprole exhibited weaker antibacterial effects against *E. faecium*, ESBL-positive *Enterobacterales*, and *A. baumannii*. Therefore, for bloodstream infection by *Staphylococci*, *E. faecalis*, ESBL-negative *Enterobacterales*, *P. aeruginosa*, and other gram-negative bacteria, ceftobiprole can be an alternative agent in clinical practice.

However, our study does have certain limitations that need to be acknowledged. Firstly, although the dataset of bloodstream infections was extensive, it was sourced from only about half of the provinces in China, with less coverage in the northwest and southern regions. This limitation might have impacted the comprehensive understanding of drug susceptibility patterns across the country. Secondly, research from other countries has indicated resistance to ceftobiprole in various STs of MRSA, such as ST22, ST5, and ST8 [40]. Hence, owing to the diverse genetic backgrounds of MRSA in different countries, the patterns of drug resistance also vary. In our study, ceftobiprole resistance was observed exclusively in the ST239 strain of MRSA, necessitating further investigations to elucidate its resistance mechanisms.

## Figures and Tables

**Figure 1 antibiotics-13-00165-f001:**
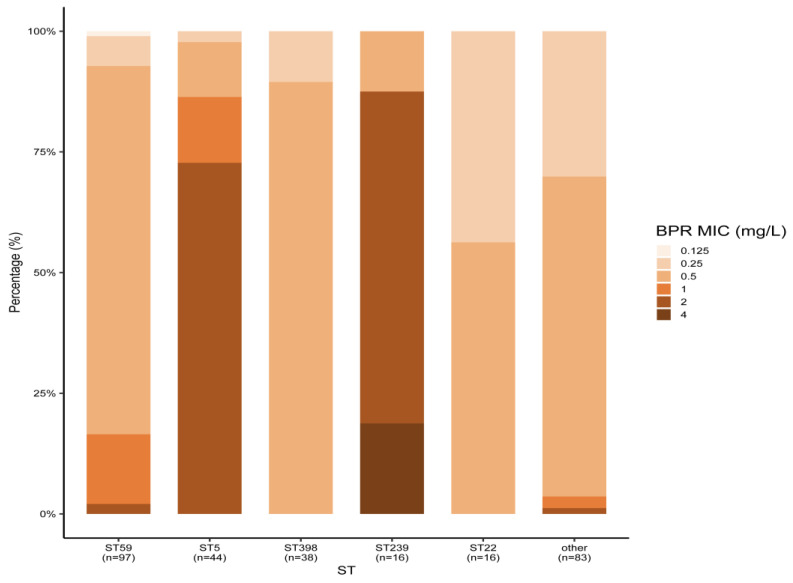
Ceftobiprole MICs of different MRSA STs isolated from blood. BPR: ceftobiprole.

**Figure 2 antibiotics-13-00165-f002:**
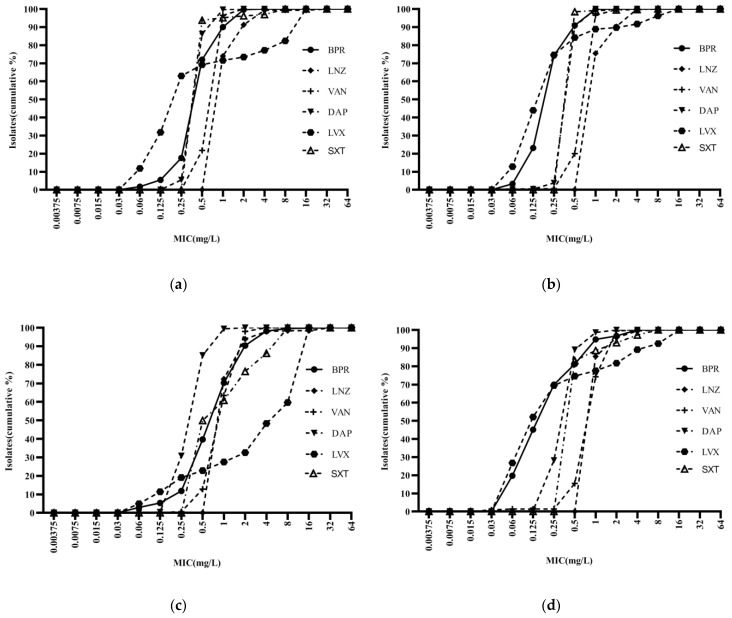
Cumulative MIC distribution of ceftobiprole and comparator compounds against Gram-positive bacteria: (**a**) MRSA (*n* = 289); (**b**) MSSA (*n* = 784); (**c**) MRCNS (*n* = 491); and (**d**) MSCNS (*n* = 213). BPR: ceftobiprole; LNZ: linezolid; VAN: vancomycin; DAP: daptomycin; LVX: levofloxacin; and SXT: cotrimoxazole.

**Figure 3 antibiotics-13-00165-f003:**
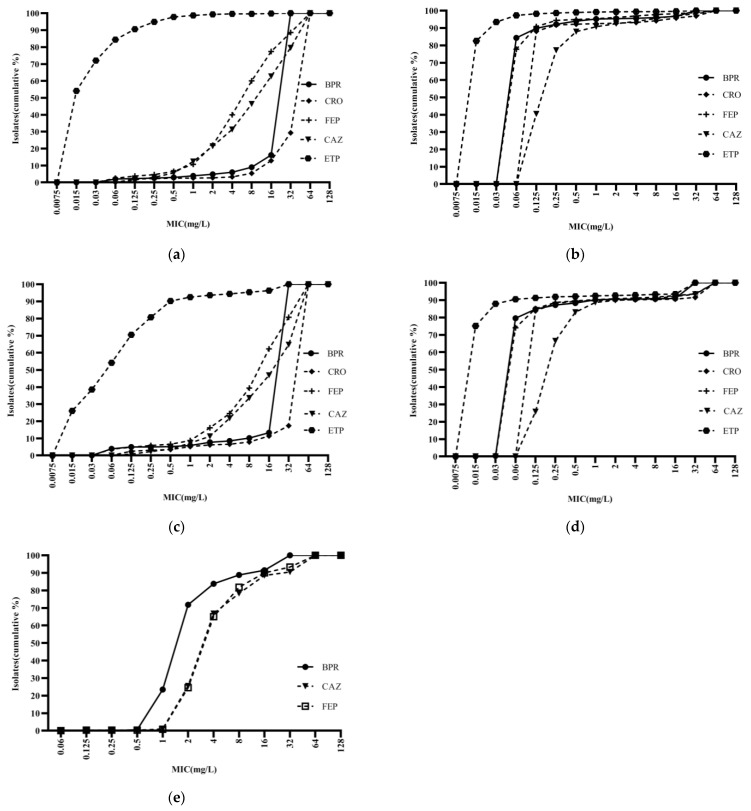
Cumulative MIC distribution of ceftobiprole and comparator compounds against Gram-negative bacilli isolated from blood: (**a**) *Escherichia coli* (ESBL+; *n* = 2035); (**b**) *Escherichia coli* (ESBL−; *n* = 2050); (**c**) *Klebsiella pneumoniae* (ESBL+; *n* = 482); (**d**) *Klebsiella pneumoniae* (ESBL−; *n* = 1404); and (**e**) *Pseudomonas aeruginosa* (*n* = 400). BPR: ceftobiprole; CRO: ceftriaxone; FEP: cefepime; CAZ: ceftazidime; and ETP: ertapenem.

**Figure 4 antibiotics-13-00165-f004:**
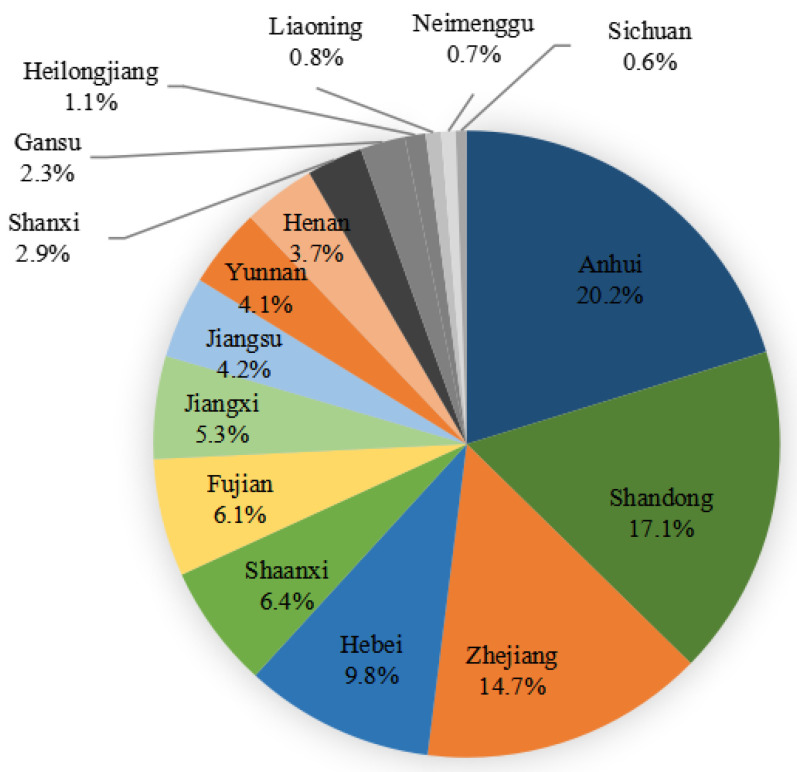
Chinese provinces distribution of 9860 bacterial strains collected from January 2021 to December 2021.

**Table 1 antibiotics-13-00165-t001:** Activities of ceftobiprole and comparator compounds against Gram-positive bacteria isolated from blood.

Agent	MIC (mg/L) ^1^			R (%) ^4^	S (%) ^5^
	MIC Range	MIC_50_ ^2^	MIC_90_ ^3^		
**Methicillin-resistant *Staphylococcus aureus* (MRSA) (289)**					
Ceftobiprole	≤0.06–4	0.5	1	0.4	99.6
Linezolid	≤1–32	≤1	2	0.7	99.3
Daptomycin	0.25–2	0.5	1	/ ^6^	99.6
Vancomycin	0.5–2	1	1	0	100.0
Cotrimoxazole	≤0.5–>8	≤0.5	≤0.5	3.8	96.2
Rifampicin	≤0.00375–>8	≤0.00375	≤0.00375	1.4	97.6
Levofloxacin	≤0.06–>16	0.25	16	26.6	71.6
**Methicillin-susceptible *Staphylococcus aureus* (MSSA) (784)**					
Ceftobiprole	≤0.06–2	0.25	1	0	100.0
Linezolid	≤1–4	≤1	2	0	100.0
Daptomycin	0.25–1	0.5	0.5	0	100.0
Vancomycin	≤0.03–4	1	1	0	99.7
Cotrimoxazole	≤0.5–8	≤0.5	≤0.5	0.5	99.5
Rifampicin	≤0.00375–>8	≤0.00375	≤0.00375	0.5	99.4
Levofloxacin	≤0.06–>16	0.25	4	10.3	88.9
**Methicillin-resistant coagulase-negative *Staphylococcus* (MRCNS) (491)**					
Ceftobiprole	≤0.06–>32	1	2	9.6	90.4
Linezolid	≤1–>32	≤1	2	2.0	98.0
Daptomycin	0.125–2	0.5	1	/	99.4
Vancomycin	≤0.03–4	1	2	0	100.0
Cotrimoxazole	≤0.5–>8	≤0.5	8	23.2	76.8
Rifampicin	≤0.00375–>8	≤0.00375	0.0075	8.4	91.6
Levofloxacin	≤0.06–>16	8	16	67.4	27.5
**Methicillin-susceptible coagulase-negative *Staphylococcus* (MSCNS) (213)**					
Ceftobiprole	≤0.06–4	0.25	1	3.3	96.7
Linezolid	≤1–4	≤1	2	0.5	99.5
Daptomycin	0.125–2	0.5	1	/	98.6
Vancomycin	≤0.03–4	1	2	0	100.0
Cotrimoxazole	≤0.5–>8	≤0.5	2	7.0	93.0
Rifampicin	≤0.00375–8	≤0.00375	≤0.00375	1.4	98.6
Levofloxacin	≤0.06–>16	0.125	8	18.3	77.5
***Enterococcus faecium* (397)**					
Ceftobiprole	0.125–>32	>32	>32	- ^7^	-
Tigecycline	≤0.06–2	0.125	0.125	-	98.5
Daptomycin	≤0.06–>8	1	2	0.5	99.5
Vancomycin	0.5–>4	1	1	0	100.0
Linezolid	≤1–>4	≤1	2	0	98.2
Teicoplanin	≤0.125–>2	0.5	0.5	0	100.0
Levofloxacin	0.25–>16	16	>16	80.9	16.9
Rifampicin	0.25–>8	4	8	77.3	11.8
Ciprofloxacin	≤0.125–>16	16	>16	88.7	9.1
Erythromycin	≤1–>32	32	>32	80.9	9.1
Penicillin G	≤1–>32	>32	>32	83.1	16.9
***Enterococcus faecalis* (296)**					
Ceftobiprole	0.125–>32	0.25	2	-	-
Tigecycline	≤0.06–0.25	0.125	0.125	0	100.0
Daptomycin	≤0.06–>8	1	2	0.7	92.9
Vancomycin	≤0.25–>2	1	2	0	100.0
Linezolid	≤1–>4	1	2	0	95.3
Teicoplanin	≤0.125–>2	0.5	0.5	0	100.0
Levofloxacin	0.25–>16	1	16	31.4	68.2
Rifampicin	0.25–>8	2	8	49.0	25.0
Ciprofloxacin	≤0.125–>16	1	>16	32.1	64.5
Erythromycin	≤1–>32	32	>32	57.1	32.1
Penicillin G	≤1–>32	2	32	13.5	86.5

^1^ MIC: minimum inhibitory concentration. ^2^ MIC_50_: minimum inhibitory concentration that inhibits 50% of bacterial isolates tested. ^3^ MIC_90_: minimum inhibitory concentration that inhibits 90% of bacterial isolates tested. ^4^ R: resistance rate (%). ^5^ S: sensitivity rate (%). ^6^ “/”: no breakpoint data for resistance of daptomycin to the corresponding bacteria. ^7^ “-”: no relevant breakpoint data were available for related drug against the corresponding bacteria.

**Table 2 antibiotics-13-00165-t002:** Activities of ceftobiprole and comparator compounds against different sequence types (STs) of MRSA isolated from blood.

Agent	MIC (mg/L) ^1^			R (%) ^4^	S (%) ^5^
	MIC Range	MIC_50_ ^2^	MIC_90_ ^3^		
**ST59 (97)**					
Ceftobiprole	0.125–2	0.5	1	0	100.0
Linezolid	≤1–2	≤1	2	0	100.0
Daptomycin	≤0.06–1	0.5	1	0	100.0
Vancomycin	0.125–4	1	1	0	99.0
Cotrimoxazole	≤0.5	≤0.5	≤0.5	0	100.0
Rifampicin	≤0.00375–>8	≤0.00375	0.0075	3.1	96.9
Levofloxacin	≤0.06–>16	0.125	4	11.3	88.7
**ST5 (44)**					
Ceftobiprole	0.25–2	2	2	0	100.0
Linezolid	≤1–2	≤1	2	0	100.0
Daptomycin	≤0.06–1	0.25	1	0	100.0
Vancomycin	0.25–2	0.5	1	0	100.0
Cotrimoxazole	≤0.5	≤0.5	≤0.5	0	100.0
Rifampicin	≤0.00375–0.015	≤0.00375	0.0075	0	100.0
Levofloxacin	0.125–>16	16	>16	84.1	15.9
**ST398 (38)**					
Ceftobiprole	0.25–0.5	0.5	0.5	0	100.0
Linezolid	≤1–2	≤1	2	0	100.0
Daptomycin	≤0.06–1	0.25	1	0	100.0
Vancomycin	0.5–2	0.5	1	0	100.0
Cotrimoxazole	≤0.5–2	≤0.5	≤0.5	0	100.0
Rifampicin	≤0.00375–0.015	≤0.00375	0.0075	0	100.0
Levofloxacin	0.125–8	0.25	0.5	2.6	97.4
**ST239 (16)**					
Ceftobiprole	0.5–4	2	4	18.8	81.2
Linezolid	≤1–2	≤1	2	0	100.0
Daptomycin	0.125–1	0.5	0.5	0	100.0
Vancomycin	0.5–2	0.5	2	0	100.0
Cotrimoxazole	≤0.5–8	≤0.5	8	18.8	81.2
Rifampicin	≤0.00375–>8	8	>8	56.3	43.7
Levofloxacin	0.125–>16	16	>16	87.5	12.5
**ST22 (16)**					
Ceftobiprole	0.25–0.5	0.5	0.5	0	100.0
Linezolid	≤1–2	≤1	2	0	100.0
Daptomycin	0.25–1	0.5	1	0	100.0
Vancomycin	0.25–1	0.5	1	0	100.0
Cotrimoxazole	≤0.5	≤0.5	≤0.5	0	100.0
Rifampicin	≤0.00375–0.0075	≤0.00375	0.0075	0	100.0
Levofloxacin	≤0.06–0.5	0.125	0.25	0	100.0
**Other (83)**					
Ceftobiprole	0.25–2	0.5	0.5	0	100.0
Linezolid	≤1–2	≤1	2	0	100.0
Daptomycin	0.125–1	0.5	1	0	100.0
Vancomycin	0.25–2	0.5	1	0	100.0
Cotrimoxazole	≤0.5–1	≤0.5	≤0.5	0	100.0
Rifampicin	≤0.00375–>8	≤0.00375	0.015	1.2	96.4
Levofloxacin	≤0.06–16	0.25	8	14.5	85.5

^1^ MIC: minimum inhibitory concentration. ^2^ MIC_50_: minimum inhibitory concentration that inhibits 50% of bacterial isolates tested. ^3^ MIC_90_: minimum inhibitory concentration that inhibits 90% of bacterial isolates tested. ^4^ R: resistance rate (%). ^5^ S: sensitivity rate (%).

**Table 3 antibiotics-13-00165-t003:** Activities of ceftobiprole and comparator compounds against Gram-negative bacilli isolated from blood.

Agent	MIC (mg/L) ^1^			R (%) ^4^	S (%) ^5^
	MIC Range	MIC_50_ ^2^	MIC_90_ ^3^		
***Escherichia coli* (ESBL−) (2050)**					
Ceftobiprole	≤0.03–>32	0.06	0.25	7.9	92.1
Ceftazidime	≤0.125–>64	0.25	1	4.5	93.8
Ceftriaxone	≤0.125–>64	≤0.125	0.25	7.2	92.5
Cefepime	≤0.06–>64	≤0.06	0.125	2.2	96.1
Ertapenem	≤0.015–>32	≤0.015	0.03	0.8	99.2
***Escherichia coli* (ESBL+) (2035)**					
Ceftobiprole	≤0.06–>32	32	>32	97.4	2.6
Ceftazidime	≤0.125–>64	16	64	53.5	31.3
Ceftriaxone	≤0.125–>64	64	>64	97.3	2.6
Cefepime	≤0.06–>128	8	64	40.0	15.5
Ertapenem	≤0.015–>32	≤0.015	0.125	1.3	97.8
***Klebsiella pneumoniae* (ESBL−) (1404)**					
Ceftobiprole	≤0.06–>32	≤0.06	1	12.8	87.2
Ceftazidime	≤0.125–>64	0.25	4	9.0	90.4
Ceftriaxone	≤0.125–>64	≤0.125	2	10.0	89.7
Cefepime	≤0.06–>64	≤0.06	1	8.3	90.8
Ertapenem	≤0.015–>32	≤0.015	0.06	7.5	92.1
***Klebsiella pneumoniae* (ESBL+) (482)**					
Ceftobiprole	≤0.06–>32	32	>32	95.0	5.0
Ceftazidime	≤0.125–>64	32	>64	66.4	21.8
Ceftriaxone	≤0.125–>64	64	>64	93.8	5.2
Cefepime	≤0.06–>128	16	64	60.6	16.2
Ertapenem	≤0.015–>32	0.06	0.5	5.6	90.0
***Enterobacter aerogenes* (ESBL−) (97)**					
Ceftobiprole	≤0.06–>32	≤0.06	0.125	4.1	95.9
Ceftazidime	≤0.125–>64	0.5	64	20.6	75.3
Ceftriaxone	≤0.125–>64	≤0.125	32	26.8	70.1
Cefepime	≤0.06–>64	≤0.06	0.5	1.0	98.0
Ertapenem	≤0.015–4	0.03	0.5	1.0	96.9
***Enterobacter aerogenes* (ESBL+) (14)**					
Ceftobiprole	≤0.06–>32	32	>32	78.6	21.4
Ceftazidime	≤1–>64	32	>64	85.7	14.3
Ceftriaxone	≤1–>64	64	>64	92.9	7.1
Cefepime	≤0.06–64	4	64	50.0	42.9
Ertapenem	≤0.015–>32	0.25	1	7.1	85.7
***Klebsiella oxytoca* (ESBL−) (91)**					
Ceftobiprole	≤0.06–>32	0.25	32	36.3	63.7
Ceftazidime	≤0.125–32	≤0.125	0.5	1.0	97.8
Ceftriaxone	≤0.125–16	≤0.125	1	6.6	90.1
Cefepime	≤0.06–64	≤0.06	0.5	1.0	97.8
Ertapenem	≤0.015–>32	≤0.015	0.03	2.2	96.7
***Klebsiella oxytoca* (ESBL+) (11)**					
Ceftobiprole	32–>32	32	>32	100.0	0
Ceftazidime	0.5–>64	16	>64	54.5	27.3
Ceftriaxone	32–>64	64	>64	100.0	0
Cefepime	≤0.06–>64	8	32	36.4	9.0
Ertapenem	≤0.015–8	0.06	2	18.2	81.8
***Enterobacter cloacae* (273)**					
Ceftobiprole	≤0.06–>32	≤0.06	32	26.0	74.0
Ceftazidime	≤0.125–>64	0.5	64	23.1	72.9
Ceftriaxone	≤0.125–>64	0.25	64	26.4	69.6
Cefepime	≤0.06–>64	≤0.06	16	11.4	81.3
Ertapenem	≤0.015–>32	≤0.015	1	8.1	89.0
***Salmonella* spp. (125)**					
Ceftobiprole	≤0.06–>32	≤0.06	8	10.4	88.8
Ceftazidime	≤0.125–>64	≤0.125	1	6.4	92.8
Ceftriaxone	≤0.125–>64	≤0.125	16	5.6	88.2
Cefepime	≤0.06–>64	≤0.06	0.5	4.0	93.6
Ertapenem	≤0.015–0.125	≤0.015	≤0.015	0	100.0
***Serratia marcescens* (100)**					
Ceftobiprole	≤0.06–>32	0.125	8	14.0	86.0
Ceftazidime	≤0.125–>64	0.5	8	6.0	91.0
Ceftriaxone	≤0.125–>64	0.25	16	8.0	88.0
Cefepime	≤0.06–64	0.125	2	4.0	93.0
Ertapenem	≤0.015–4	≤0.015	0.25	5.0	93.0
***Pseudomonas aeruginosa* (400)**					
Ceftobiprole	0.5–>32	2	16	- ^6^	-
Ceftazidime	1–>64	4	32	11.5	78.3
Cefepime	0.125–>64	4	16	10.0	81.8
***Acinetobacter baumannii* (308)**					
Ceftobiprole	≤0.06–>32	32	>32	-	-
Ceftazidime	≤0.125–>64	64	>64	61.4	37.0
Cefepime	≤0.06–>64	64	>64	63.3	35.4

^1^ MIC: minimum inhibitory concentration. ^2^ MIC_50_: minimum inhibitory concentration that inhibits 50% of bacterial isolates tested. ^3^ MIC_90_: minimum inhibitory concentration that inhibits 90% of bacterial isolates tested. ^4^ R: resistance rate (%). ^5^ S: sensitivity rate (%). ^6^ “-”: no relevant breakpoint data were available for ceftobiprole against the corresponding bacteria.

## Data Availability

The data presented in this study are available upon request.

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
