# Peer review of "In Vitro Antibacterial Activity of Ceftobiprole and Comparator Compounds against Nation-Wide Bloodstream Isolates and Different Sequence Types of MRSA"

_antibiotics, 2024, doi:10.3390/antibiotics13020165_

Round 1
Reviewer 1 Report
Comments and Suggestions for Authors
The manuscript provides a comprehensive investigation into the in vitro antibacterial activity of ceftobiprole against a diverse range of bacterial strains isolated from bloodstream infections in China. The study addresses an important clinical concern given the rising cases of sepsis and the role of antibiotic resistance in these infections. The results are presented clearly, and the discussion interprets the findings effectively. However, there are areas that need clarification and improvement to enhance the manuscript's scientific rigor and clarity.
Major Comments:
1. The manuscript briefly mentions resistance in MRSA strains but does not delve into the potential mechanisms. Providing insights into the genetic basis of resistance, especially in ST239 MRSA strains, and discussing how these findings align with existing literature would strengthen the discussion section.
2. The manuscript could be improved by clearly discussing the clinical implications of the findings. How do the in vitro results translate to potential therapeutic strategies or challenges in the clinical setting?
3. The conclusion section can be strengthened by summarizing the key findings and their implications in a concise manner. Additionally, consider including potential limitations of the study.
4. The manuscript could benefit from additional recent references to provide readers with the latest context on sepsis and antibiotic resistance.
Minor Comments:
1. Introduce what ESBL stand for at the beginning of the introduction.
2. Change "These results provide important information clinicians" to "These results provide important information to clinicians" in the Abstract.
Reviewer 2 Report
Comments and Suggestions for Authors
The work raises the significant issue of sensitivity of antibiotics from the beta-lactam group. Through their work, scientists have shown that ceftobiprole can be an alternative in treating patients. The study was carried out on the basis of a massive group of microorganisms. However, there is no statistical analysis that would comprehensively take into account all the obtained results. Furthermore, it is worth considering that the subtitles already indicate the result obtained (they now indicate what was done). The work was carried out in accordance with European standards, including EUCAST.
The charts could be described in more detail, which would introduce the reader better to the topic under study.
The manuscript requires minor language editing.
Reviewer 3 Report
Comments and Suggestions for Authors
This is a straight forward in vitro susceptibility study of ceftobiprole, but the reader does not know this from the title or the introduction where the emphasis seems to be on MRSA. The sequence typing of MRSA could even be a separate study since this sub study is not clearly stated. The reader does not know there were nearly 10,000 isolates, both Gram positive and Gram negative. In fact MRSA were a relatively small number.
Any multicenter study needs to clearly identify how the clinical isolates were selected. Did they include all consecutive BSI isolates, were they from unique patients and the methods of the AST was not identified, neither was the critical fact of what laboratory or laboratories performed the AST. A central lab, multiple regional or hospital-based labs?
The value is primarily the description in the in vitro activity of ceftobiprole in China and even though the study included > 50 sites, the geographic distribution of these sites is not described. China is a big country.
Comments on the Quality of English LanguageEnglish is quite good although I think the proper term is Enterobacterales which replace the term Enterobacteriaceae. to describe glucose fermenting Gram negative aerobes.
Round 2
Reviewer 3 Report
Comments and Suggestions for Authors
The revisions to the original manuscript addressed my concerns. The new references are well selected and the discussion expanded to address further questions. I do have three minor comments/edits:
1. line 20 of abstract. The transition from discussing Gram-positives to Gram negatives is not correct English. The word thus is in reference to the previous sentences which is about sequence typing of MRSA and has no relationship to the ensuing discussion of Enterobacterales and non-fermenters. I suggest a new paragraph and starting with Among Gram negative isolates tested.
2. It still is not clear until late in the paper that the MRSA isolates tested for ST were a totally separate collection during a different time period. This should be made clear earlier, e.g. line 172
3. line 290, Staphylococcus aureus should be italicized.
